# Effects of Chronic High-Frequency rTMS Protocol on Respiratory Neuroplasticity Following C2 Spinal Cord Hemisection in Rats

**DOI:** 10.3390/biology11030473

**Published:** 2022-03-19

**Authors:** Pauline Michel-Flutot, Isley Jesus, Valentin Vanhee, Camille H. Bourcier, Laila Emam, Abderrahim Ouguerroudj, Kun-Ze Lee, Lyandysha V. Zholudeva, Michael A. Lane, Arnaud Mansart, Marcel Bonay, Stéphane Vinit

**Affiliations:** 1Université Paris-Saclay, UVSQ, Inserm, END-ICAP, 78000 Versailles, France; pauline.michel78280@yahoo.fr (P.M.-F.); isleyj@yahoo.com.br (I.J.); valentin.vanhee.pro@gmail.com (V.V.); camille.bourcier@uvsq.fr (C.H.B.); ouguerroudj.a@gmail.com (A.O.); marcel.bonay@bch.aphp.fr (M.B.); 2Université Paris-Saclay, UVSQ, Inserm, Infection et Inflammation (2I), 78000 Versailles, France; laila.emam@uvsq.fr (L.E.); arnaud.mansart@uvsq.fr (A.M.); 3Department of Biological Sciences, National Sun Yat-sen University, Kaohsiung 80424, Taiwan; kzlee@mail.nsysu.edu.tw; 4Gladstone Institutes, San Francisco, CA 94158, USA; lvzholudeva@gmail.com; 5Marion Murray Spinal Cord Research Center, Department of Neurobiology and Anatomy, Drexel University College of Medicine, Philadelphia, PA 19129, USA; mlane.neuro@gmail.com

**Keywords:** spinal cord injury, repetitive transcranial magnetic stimulation, phrenic motor network, neuroplasticity, motoneuron excitability, diaphragm muscle

## Abstract

**Simple Summary:**

High spinal cord injuries (SCIs) are known to lead to permanent diaphragmatic paralysis, and to induce deleterious post-traumatic inflammatory processes following cervical spinal cord injury. We used a noninvasive therapeutic tool (repetitive transcranial magnetic stimulation (rTMS)), to harness plasticity in spared descending respiratory circuit and reduce the inflammatory processes. Briefly, the results obtained in this present study suggest that chronic high-frequency rTMS can ameliorate respiratory dysfunction and elicit neuronal plasticity with a reduction in deleterious post-traumatic inflammatory processes in the cervical spinal cord post-SCI. Thus, this therapeutic tool could be adopted and/or combined with other therapeutic interventions in order to further enhance beneficial outcomes.

**Abstract:**

High spinal cord injuries (SCIs) lead to permanent diaphragmatic paralysis. The search for therapeutics to induce functional motor recovery is essential. One promising noninvasive therapeutic tool that could harness plasticity in a spared descending respiratory circuit is repetitive transcranial magnetic stimulation (rTMS). Here, we tested the effect of chronic high-frequency (10 Hz) rTMS above the cortical areas in C2 hemisected rats when applied for 7 days, 1 month, or 2 months. An increase in intact hemidiaphragm electromyogram (EMG) activity and excitability (diaphragm motor evoked potentials) was observed after 1 month of rTMS application. Interestingly, despite no real functional effects of rTMS treatment on the injured hemidiaphragm activity during eupnea, 2 months of rTMS treatment strengthened the existing crossed phrenic pathways, allowing the injured hemidiaphragm to increase its activity during the respiratory challenge (i.e., asphyxia). This effect could be explained by a strengthening of respiratory descending fibers in the ventrolateral funiculi (an increase in GAP-43 positive fibers), sustained by a reduction in inflammation in the C1–C3 spinal cord (reduction in CD68 and Iba1 labeling), and acceleration of intracellular plasticity processes in phrenic motoneurons after chronic rTMS treatment. These results suggest that chronic high-frequency rTMS can ameliorate respiratory dysfunction and elicit neuronal plasticity with a reduction in deleterious post-traumatic inflammatory processes in the cervical spinal cord post-SCI. Thus, this therapeutic tool could be adopted and/or combined with other therapeutic interventions in order to further enhance beneficial outcomes.

## 1. Introduction

High spinal cord injuries (SCIs) induce long-lasting neuromotor deficits, such as respiratory insufficiency [1]. Patients living with such injuries often rely on ventilatory assistance to survive, although some can be weaned off with time, exemplifying spontaneous plasticity. The rodent C2 hemisection (C2HS) model is one of the most common preclinical models to study respiratory system neuroplasticity and neuroinflammation. A C2HS disrupts descending input to ipsilateral phrenic motoneurons that innervate the diaphragm, the main inspiratory muscle, thus resulting in diaphragm hemiplegia [2,3,4,5,6,7,8,9,10]. The contralateral side remains intact, allowing the animal to survive.

Limited spontaneous recovery of the injured hemidiaphragm activity is observed in this model of SCI, characterized by a partial reactivation of phrenic motor networks and diaphragm activities. This reactivation is sustained by normally silent respiratory pathways crossing the spinal midline at the C3–C6 spinal cord levels, called the crossed phrenic phenomenon (CPP) [11,12,13,14]. However, this marginal spontaneous plasticity is too weak to contribute to significant ventilatory recovery following C2HS [15]. Strengthening the CPP and providing new intraspinal connections to the denervated motoneurons is a potential target for developing novel therapeutic tools in order to further improve respiratory function following high SCI.

A noninvasive approach to stimulate neural activity is transcranial magnetic stimulation (TMS), which involves applying a high output magnetic field above the neuronal areas. In fact, TMS applied as a single pulse or in a repetitive way (rTMS) is a noninvasive and painless method already used in the clinic to diagnose and treat many disorders [16,17,18,19,20], as well as a potential therapeutic tool in preclinical models of cognitive impairment [21,22]. This technique operates through its neuromodulatory effects on neuronal circuitry [23,24]. The potential of rTMS to improve outcomes following incomplete SCI has gained recognition in the past few years but has mainly been applied to enhance locomotor recovery [25,26,27] and sensorimotor restoration [27,28,29] in preclinical models of SCI. 

We recently demonstrated that a single train of TMS delivered above the animals’ motor threshold can induce a long-lasting increase in phrenic system excitability, as measured with diaphragm motor evoked potentials (MEPdia) [30]. While this study was focused on determining stimulation parameters in naïve, anesthetized rats, it demonstrated that MEPdia can be used as a reliable and reproducible technique for assessing phrenic system excitability during TMS [31,32,33,34]. While there is interest in the therapeutic potential of rTMS following SCI among researchers, little is known about the cellular and molecular mechanisms that sustain the neuromodulatory effects of acute or chronic rTMS. A few in vivo and in vitro studies have been conducted to elucidate potential cellular mechanisms [35,36,37,38], including excitatory neurotransmission via N-methyl-D-aspartic acid (NMDA) and α-amino-3-hydroxy-5-méthylisoxazol-4-propionate (AMPA) (GluR1 subunit) receptor pathways [37,39,40], and inhibitory neurotransmission via γ-aminobutyric acid (GABA) system [41,42]. For example, rTMS protocols can modulate the expression of neuronal activity markers such as c-fos. Low-frequency repetitive magnetic stimulation (rMS) has been shown to increase nuclear, neuronal c-fos expression in rat organotypic cortex brain slices [43], whereas rTMS theta-burst stimulation resulted in decreased neuronal c-fos expression [38]. In addition, repetitive magnetic stimulation (rMS) protocol on SH-SY5Y neuroblastoma cells induced an increase in cAMP and phospho-CREB expression [44]. In vivo, protocols using high-frequency repetitive trans-spinal magnetic stimulation (rTSMS) also resulted in reduced expression of markers for apoptosis and neuronal death, while the expression of markers of axonal growth and neuronal proliferation were upregulated. These results were accompanied by reduced axonal demyelination [45]. A deeper understanding of rTMS-regulated molecular signaling pathways following SCI could, therefore, help to harness the potential beneficial effects of rTMS as a therapeutic intervention.

The putative effects of rTMS on neuroinflammation are also of great interest. A few studies have used rMS on glial cells, but its effects are diverse and depend on the stimulation parameters used and the model studied [46]. Chronic low-frequency rTMS used on naive rats did not induce observable changes in astrocyte and microglial density, supporting the safety of this protocol regarding glial cell homeostasis in the normal/control condition [47]. However, conflicting results have been observed in in vivo studies. For instance, chronic high-frequency rTMS increased astrocytic and microglial density in a preclinical model of ischemia in gerbils (hippocampus) [48], whereas a decrease in cellular density in a preclinical model of T9 dorsal SCI (compression) [49]. In addition, some studies also showed a beneficial neuroinflammatory effect, observed through the decreased release of TNFα (proinflammatory cytokine) in substantia nigra in a model of Parkinson’s disease [50,51].

To our knowledge, there have been no studies investigating the potential therapeutic effects of high-frequency rTMS on impaired respiratory function following cervical SCI, specifically at the phrenic circuit level. The present study aimed to test the hypothesis that chronic 10 Hz rTMS can improve respiratory function after SCI. Here, we test this hypothesis using cellular, molecular, and electrophysiological outcome measures to assess the potential therapeutic benefits of rTMS in a preclinical model of C2HS in adult, Sprague Dawley rats. 

## 2. Materials and Methods

### 2.1. Ethics Statement

Adult Sprague Dawley male rats (Janvier, France; n = 41, 350–450 g) were used for this study. Experiments were approved by the Ethics Committee of the University of Versailles Saint-Quentin-en-Yvelines and complied with the French and European laws (EU Directive 2010/63/EU) regarding animal experimentation (Apafis #2017111516297308_v3). 

Animals were dually housed in ventilated cages in a state-of-the-art animal care facility (2CARE animal facility, accreditation A78-322-3, France) on a 12 h light–dark cycle, with access to food and water ad libitum. 

### 2.2. Chronic C2 Hemisection

#### 2.2.1. Intrapleural CTB Injection and Surgery

Prior to anesthesia, animals were premedicated subcutaneously with buprenorphine (Buprécare, 0.03 mg/kg), trimethoprim, and sulfadoxine (Borgal 24%, 30 mg/kg), medetomidine (Médétor, 0.1 mg/kg) and carprofen (Rimadyl, 5 mg/kg). 10 min after the injections, animals were anesthetized with isoflurane (5% in 100% O_2_) in a closed chamber. Rats were then intubated and ventilated with a rodent ventilator (model 683; Harvard Apparatus, South Natick, MA, USA), and anesthesia was maintained throughout the surgical procedure with isoflurane (2.5% in 100% O_2_). For phrenic motoneuron retrograde labeling, intrapleural injections of cholera toxin B fragment were performed bilaterally in all animals (15 µL/side) using a custom needle (6 mm, 23 gauge, semi-blunt to avoid puncturing of the lung) and a 50 μL Hamilton syringe as described previously [52]. After skin and muscles were retracted, laminectomy and durotomy were performed at the C2 level. The spinal cord was then sectioned unilaterally (left side) with microscissors. To ensure the section of potentially remaining fibers, a microscalpel was used immediately after microscissors, as described previously [7].

#### 2.2.2. Brainstem Neuronal Retrograde Labeling with Hydroxystilbamidine

For respiratory brainstem neuronal retrograde labeling, a sterilized piece of cotton was impregnated with 2 µL of 7% hydroxystilbamidine (Fluorogold) and left into the injury site for 20 min. Then, the lesion site was flushed with sterilized saline, and muscles and skin were then sutured closed. To reverse medetomidine-induced anesthesia, atipamezole (Revertor, 0.5 mg/kg) was intramuscularly injected. Isoflurane anesthesia was then turned off, and the endotracheal tube was removed when animals showed signs of wakefulness. All animals were kept 7 days postsurgery in their cage to recover before rTMS or Sham rTMS protocol was applied.

### 2.3. Repetitive TMS (rTMS) Protocol

rTMS protocol was performed using the magnetic stimulator MAGPRO R30 (Magventure, Farum, Denmark) connected to a figure-of-eight coil (Cool-B65), delivering a unique biphasic pulse with the intensity of the stimulus expressed as a percentage of a maximum output of the stimulator (% MO). The protocol (9 trains of 100 biphasic pulses, separated by 30 s intervals between trains delivered at 50% MO, 900 stimulations per protocol) was applied in awake restrained animals at −6 mm caudal to Bregma. This protocol induced a long-lasting increase in phrenic excitability in anesthetized, intact rats [30]. Control animals received a Sham rTMS protocol (e.g., no stimulation but the same time spent in the custom-designed restraining device, Figure 1). This rTMS protocol was applied 7 days postinjury for either 7 days (once a day), 1 month, or 2 months (once a day, 5 days per week) (Figure 1).

### 2.4. Electrophysiological Recordings

#### 2.4.1. Animal Preparation

Animals were randomly divided into 6 groups: 7-day Sham rTMS (*n* = 8); 7-day 10 Hz rTMS (*n* = 9); 1-month Sham rTMS (*n* = 6); 1-month 10 Hz rTMS (*n* = 6); 2-month Sham rTMS (*n* = 6); 2-month 10 Hz rTMS (*n* = 6). Electrophysiological recordings of diaphragm activity and excitability were used to functionally evaluate the effects of rTMS treatment on the phrenic motor circuit after completion of sham or 10 Hz rTMS. Briefly, anesthesia was induced using isoflurane (5% in 21% O_2_ balanced) in an anesthesia chamber and maintained through a nose cone (2.5% in 100% air balanced). Animals were tracheotomized and pump-ventilated (Rodent Ventilator, model 683; Harvard Apparatus, South Natick, MA, USA). The ventilation rate (frequency > 72 breaths per minute, tidal volume: 2.5 mL) was adjusted to reduce the end-tidal CO_2_ value below the animals’ central apneic threshold throughout the experiment to avoid recording spontaneous diaphragm contractions. During the recordings, animals were placed on a heating pad to maintain a constant body temperature (37.5 ± 0.5 °C), and their rectal temperature was continuously monitored throughout the experiment. Arterial pressure was measured through a catheter inserted into the right femoral artery. Arterial and tracheal pressures were monitored continuously with transducers connected to a bridge amplifier (AD Instruments, Dunedin, New Zealand). The depth of anesthesia was confirmed by the absence of response to toe pinch. A laparotomy was performed, and the liver was gently moved dorsally to access the diaphragm. Gauze soaked with warm phosphate-buffered saline was placed on the liver to prevent dehydration. Both sides (ipsilateral and contralateral to the spinal cord lesion) of the diaphragm were implanted with two custom-made hooked bipolar electrodes into each mid-costal part of the diaphragm and left in place for the duration of the experiment for the measurement of (1) spontaneous diaphragm EMG during spontaneous poïkilocapnic normoxic or transient mild asphyxia breathing (by occlusion of the animal’s nose for 15 s after disconnection of a tracheal tube from the ventilator) and (2) diaphragm MEP when PETCO2 was below the apneic threshold. 

#### 2.4.2. Diaphragmatic EMG Recordings

EMGs were amplified (Model 1800; gain, 100; A-M Systems, Everett, WA, USA) and band pass-filtered (100 Hz to 10 kHz). The signals were digitized with an 8-channel Powerlab data acquisition device (Acquisition rate: 4 k/s; AD Instruments, Dunedin, New Zealand), connected to a computer, and analyzed using LabChart 7 Pro software (AD Instruments, Dunedin, New Zealand). The bilateral diaphragmatic EMGs were integrated (50 ms decay).

#### 2.4.3. Diaphragmatic MEP Recordings

Next, the head of the animal was placed on a nonmagnetic, custom-made stereotaxic apparatus, which allowed its positioning from the center of the figure-of-eight coil to −6 mm from Bregma, at an angle of 0°, as previously described [30,31]. MEPdia induced by a single pulse of TMS was recorded (summation of 5 to 10 TMS pulses between 2 heartbeats, max 10 trials at 90% MO). These electromyographic signals were amplified (gain, 1 k; A-M Systems, Everett, WA, USA) and band pass-filtered (100 Hz to 10 kHz). The signals were then digitized with an 8-channel Powerlab data acquisition device (Acquisition rate: 100 k/s; AD Instruments, Dunedin, New Zealand) connected to a computer and analyzed using LabChart 8 Pro software (AD Instruments, Dunedin, New Zealand).

### 2.5. Tissue Processing

At the end of the experiment, animals were euthanized by intracardiac injection of pentobarbital (EXAGON, Axience), intracardially perfused with heparinized 0.9% NaCl (10 mL), followed by Antigenfix solution (DIAPATH). After perfusion, the C1–C6 spinal cord and brainstem were carefully dissected and stored at 4 °C in fixative for 24 h. After postfixation, tissues were cryoprotected for 48 h in 30% sucrose (in 0.9% NaCl), and stored at −80 °C. Frozen longitudinal (C1–C3 spinal cord) and transverse (C3–C6 spinal cord and brainstem) free-floating sections (30 µm) were cut using a Thermo Fisher cryostat. C1–C3, C3–C6, and brainstem sections were stored in a cryoprotectant solution (Sucrose 30%, ethylene glycol 30%, and PVP40 1% in PBS 1×) at −22 °C. Every fifth section from C1–C3 was used for lesion reconstruction to examine the extent of C2 injury using cresyl violet histochemistry.

### 2.6. Histological Reconstruction of the Extent of C2 Injury

Longitudinal sections from the C1–C3 cord were used to assess the dorsoventral and mediolateral extent of injury in all animals. Brightfield microscopy was used to examine the cresyl violet-stained sections and recorded on a stereotaxic transverse plane of the C2 spinal cord. Each injury was then digitized and analyzed with ImageJ software (NIH). The extent of the injury on the injured side was calculated using a reference to a complete hemisection (which is 100% of the hemicord) and reported as a percentage (Figure 2), as described in our previous publication [7].

### 2.7. Immunofluorescence

For immunofluorescence experiments, free-floating transverse sections of the C3–C6 spinal cord and brainstem were washed and placed in blocking solution (NDS 5% in PBS 1×) for 30 min and then incubated with the corresponding antibody in blocking solution (NDS 5%) overnight on an orbital shaker at 4 °C. After several PBS washes, sections were incubated in the corresponding secondary antibody for 2 h at room temperature, then washed again with PBS. The following primary antibodies were used: cholera toxin, B-subunit (CTB, Calbiochem, Saint-Quentin-Fallavier, France, 1/1000, goat polyclonal), CREB (Sigma, Saint-Quentin-Fallavier, France, 1/2000, rabbit polyclonal), GAP-43 (Sigma, Saint-Quentin-Fallavier, France, 1/2000, mouse monoclonal), nitric oxide synthase II (iNOS, Millipore-Merck, Guyancourt, France, 1/3000, rabbit polyclonal), CD68 (Millipore-Merck, Guyancourt, France, 1/300, mouse monoclonal), Iba1 (Abcam, Paris, France, 1/400, goat polyclonal), phospho-c-Jun (Ser63) II (Cell Signaling, Saint-Cyr-L’Ecole, France, 1/200, rabbit polyclonal), and GFAP (Millipore-Merck, Guyancourt, France, 1/4000, rabbit polyclonal). The secondary antibodies were linked to the fluorochromes Alexa Fluor 488, 594 (Molecular Probes, Illkirch, France, 1/2000) or 647 (Invitrogen, Illkirch, France, 1/2000). Biotinylated wisteria floribunda lectin (WFA, Vector laboratories, Les Ulis, France, 1/2000) with Alexa Fluor 488 Avidin (Molecular Probes, Illkirch, France, 1/1000) were used to labeled chondroitin sulfate proteoglycans (CSPGs). Images of the different sections were captured with a Hamamatsu ORCA-R² camera mounted on an Olympus IX83 P2ZF microscope or a 3dhistech panoramic slide scanner. Images were analyzed using ImageJ 1.53n software (NIH, USA).

### 2.8. Data Processing and Statistical Analyses

The amplitude (normalized to the corresponding sham group in arbitrary units, AU) of at least 5 double-integrated diaphragm EMG inspiratory bursts during normoxia and mild asphyxia was calculated for each animal from the injured and the intact sides with LabChart 7 Pro software (AD Instruments). Diaphragm MEP traces for each side (at least 5 MEPdia) were averaged and superimposed using LabChart Pro software (AD Instruments). The baseline-to-peak amplitude of the first wave of each superimposed MEPdia was calculated. 

One-way ANOVA was performed between different groups for the extent of injury evaluation. Comparisons between intact and injured sides for diaphragmatic EMG and MEP and between eupnea and asphyxia for diaphragmatic EMG were performed by Student’s paired *t*-test. Two-way ANOVA (Fisher LSD Method for multiple comparisons) was used to compare MEPdia throughout the experiment and between different rTMS protocols. Student’s *t*-tests were used to compare values of the same side (intact or injured) between the different protocols (between 7-day, 1-month, and 2-month Sham rTMS or between 7-day, 1-month, and 2-month 10 Hz rTMS) and to compare Sham and 10 Hz rTMS group values for the same time point (at 7 days, 1 month, and 2 months). Paired *t*-tests were used to compare data from intact and injured sides of the same animal.

All data are presented as mean ± SD, and statistics were considered significant when *p* < 0.05. SigmaPlot 12.5 software was used for all analyses.

## 3. Results

### 3.1. rTMS-Induced Effects on Diaphragm Activity during Eupnea

Diaphragm activity was assessed by recording EMGdia amplitude for both intact and injured sides (Figure 3A). A reduction in EMGdia amplitude was observed 15 days postinjury (P.I.) for the injured side (7-day Sham rTMS = 0.06 ± 0.08 µV.s.s), compared with intact side (0.54 ± 0.20 µV.s.s, *p* < 0.001). This reduced diaphragm activity persisted for the injured side in Sham rTMS-treated animals at 36 days P.I. (0.00 ± 0.00 µV.s.s) and 64 days P.I. (0.03 ± 0.04 µV.s.s) (*p* > 0.05). The 10 Hz rTMS protocol did not induce a significant change in EMGdia amplitude on the side of the injury at any experimental time point (7 days: 0.01 ± 0.02 µV.s.s; 1-month rTMS: 0.01 ± 0.03 µV.s.s; 2-month rTMS: 0.03 ± 0.03 µV.s.s). For the intact side, no differences in diaphragm activity were observed between Sham rTMS group (0.54 ± 0.20 µV.s.s) and 10 Hz rTMS group (0.51 ± 0.21 µV.s.s) following 7 days of stimulation (*p* = 0.772). However, following 1 month of rTMS, 10 Hz treated animals (0.63 ± 0.15 µV.s.s) presented a significantly higher EMGdia amplitude, compared with Sham-treated animals (0.45 ± 0.10 µV.s.s, p = 0.011). After 2 months of rTMS, this difference between Sham-treated animals (0.67 ± 0.14 µV.s.s) and 10 Hz treated animals (0.64 ± 0.12 µV.s.s) disappeared (*p* = 0.656) due to an increase in EMGdia amplitude between 1 month (0.45 ± 0.10 µV.s.s) and 2 months (0.67 ± 0.14 µV.s.s) in the Sham group (*p* = 0.039) (Figure 3B).

### 3.2. rTMS-Induced Effects on Diaphragm Muscle Response to Respiratory Challenge

In addition to analyzing diaphragm activity during eupneic breathing, muscle activity was also analyzed during respiratory challenges (i.e., mild asphyxia). When challenged with mild asphyxia, there was no significant change in EMGdia amplitude, compared with eupneic breathing on the intact side at 7 days in Sham-treated animals, and 7-day and 1-month 10 Hz rTMS groups (Figure 4A,B,D, respectively). EMGdia on the intact side, however, significantly decreased with mild asphyxia challenge, compared to eupneic breathing, in 1- and 2-month Sham-treated groups, as well as the 2-month 10 Hz rTMS group (Figure 4C,E,F, respectively). In contrast, EMGdia on the injured side significantly increased with 10 Hz rTMS 2 months poststimulation (Figure 4F). No other statistically significant differences were observed.

### 3.3. rTMS-Induced Effects on Phrenic System Excitability

MEPdia amplitudes were measured in response to a single pulse of TMS to evaluate phrenic excitability following rTMS protocols (Figure 5A). Significant differences in response were seen only between Sham-treated animals on the intact side from 1 to 2 months postinjury (MEPdia increased), and between intact and injured sides of Sham-treated animals at 1 month, with the injured side being significantly greater (Figure 5B). No other statistically significant differences were observed. 

### 3.4. rTMS-Induced Effects on Plasticity Markers in C3–C6 Spinal Cord

The expression of plasticity markers CREB and GAP-43 was evaluated at the C3–C6 spinal cord, the anatomical location of the phrenic motoneuron nucleus (Figure 6A). The percentage of cholera toxin beta (CTB, Figure 6A)-labeled phrenic motoneurons expressing CREB (Appendix A) was significantly reduced for the intact side after 1 month of Sham or 10 Hz rTMS. In contrast, a reduction in CREB expression was seen on the injured side only in those animals treated with 10 Hz rTMS for 1 month (Figure 6B).

GAP-43 is normally synthesized in axonal growth cones; therefore, changes in GAP-43 immunofluorescence were evaluated in the phrenic motoneuron area in the C3–C6 spinal cord (Appendix A and Figure 7A). The area labeled by antibodies against GAP-43 increased on the side of the injury 1 month after 10 Hz rTMS. No other significant differences were observed at any experimental time point on the intact or injured side (Figure 7B). Although there was significantly less GAP-43 labeling on the injured side, compared with the intact side in the ventrolateral funiculi of the spinal cord, no significant differences were seen between Sham- or 10 Hz treated animals across any of the time points exampled (Appendix A). 

There was no difference in the expression of iNOS, which produces the reactive oxygen species nitric oxide, in phrenic motoneurons despite their denervation in any of the groups (Appendix A). There was also no difference in p-c-Jun and CREB expression in Fluorogold positive identified rostral ventral respiratory group (rVRG) neurons on the side of the injury when compared between 1 and 2 months of Sham or 10 Hz rTMS treatment (Appendix A, respectively), nor were there differences in CSPG expression in the rVRG (Appendix A).

### 3.5. rTMS-Induced Effects on Neuroinflammation 

Immunohistochemistry against Iba1 (microglia) and CD68 (macrophages) was used to evaluate neuroinflammation in the injured C1–C3 spinal cord following 1 month and 2 months of rTMS treatment (Appendix A). For the injured side, the area occupied by Iba1 labeling was reduced in 1-month 10 Hz rTMS-treated animals (10.72 ± 5.73%), compared with 1-month Sham rTMS-treated animals (23.74 ± 8.45%; *p* < 0.05) between −345 µm and +345 µm from lesion epicenter, as well as between −690 µm and −345 µm from lesion epicenter (1-month 10 Hz rTMS: 13.72 ± 9.42% vs. 1-month Sham rTMS: 24.46 ± 6.31%; *p* < 0.05) (Appendix A). No difference in CD68 labeling on the side of the injury was observed between groups (Appendix A). For the intact side, there was no difference in Iba1 labeling between groups (Appendix A). However, the area occupied by CD68 positive cells was reduced in the 2-month 10 Hz rTMS group (0.02 ± 0.03%), compared with 2-month Sham rTMS (0.33 ± 0.39%; *p* < 0.05) between −345 µm and +345 µm from lesion epicenter (Appendix A).

Immunohistochemistry against GFAP (astrocytes) and WFA (CSPGs) was used to evaluate any rTMS-induced changes in the astroglial border of the lesion (Appendix A). No difference in GFAP immunofluorescence was detected on either the injured or the intact sides (Appendix A). WFA labeling was reduced in 2-month 10 Hz rTMS (9.04 ± 3.42%), compared with 1-month 10 Hz rTMS-treated animals (17.14 ± 9.54%; *p* < 0.05) on the side of the injury (Appendix A). No differences were observed in WFA immunofluorescence on the intact side (Appendix A). 

## 4. Discussion

The present study is the first to investigate the therapeutic effects of chronic rTMS for enhancing respiratory function following cervical spinal cord injury (SCI) in a preclinical model. Our previous study already demonstrated that a single acute delivery of 10 Hz rTMS in anesthetized rats induced an increase in phrenic network excitability [30]. This 10 Hz magnetic stimulation is recognized for its long-term potentiation (LTP)-like effect [42,52]. We, therefore, hypothesized that chronic delivery of this protocol would induce beneficial effects on respiratory recovery after cervical SCI.

Contrary to our expectations, the analysis of EMGdia recordings showed the application of chronic 10 Hz rTMS had no effect on diaphragm activity on the side of the injury during eupnea, regardless of the duration of treatment. Conversely, an increase in diaphragm activity on the intact side after 1 month of treatment in injured animals was observed. Nontreated animals reached an EMGdia amplitude similar to those of treated animals at 9 weeks postinjury. These results might suggest that chronic 10 Hz rTMS strengthens spared descending respiratory pathways, reflecting a recovery plateau. Indeed, between 4 and 8 weeks post-C2 hemisection, a plateau in diaphragm activity has been observed when treated with intermittent hypoxia [53]. A similar observation was also found when a chronic protocol of intermittent hypoxia was applied following C2 hemisection in rats on diaphragm activity, with no difference between treated animals and normoxic animals after 3 weeks of treatment (spontaneous recovery reaches a ceiling/plateau by that time postinjury) [54]. Additionally, consistent with MEPdia results, the intact hemidiaphragm could better compensate for the paralyzed hemidiaphragm. Indeed, animals treated with 10 Hz rTMS did not differ significantly in MEP amplitude over time in either intact or injured hemidiaphragm. In contrast, Sham-treated animals had a reduced amplitude for the intact side, compared with the injured side following 1 month of treatment, and similar to those of rTMS-treated rats after 2 months of treatment. These results suggest that excitability of spared phrenic motoneurons is reduced in Sham-treated animals, whereas rTMS treatment maintained basal excitability in these motoneurons. Moreover, at 2 months post-rTMS treatment, an increased response to asphyxia was observed on the injured side, whereas no response was observed in Sham-treated animals. This reflects a strengthening of the existing CPP, which is consistent with evidence for spontaneous CPP 3 months post-C2 hemisection but not seen at 7 days postinjury [5].

The reduction in CREB expression in phrenic motoneurons occurred earlier postinjury in treated animals, suggesting that CREB signaling may contribute to treatment-driven plasticity. Moreover, the reduction in CD68 positive cells on the intact side of the C1–C3 spinal cord could reflect the anti-inflammatory effect of rTMS treatment, which may also contribute to neuroplasticity. Consistent with this finding, others have shown a reduction in GFAP and Iba1 labeling, which correlated with increased neuronal plasticity after rTSMS treatment [45,55]. Although these effects correlate with neuroplastic processes, no change in CREB or p-c-Jun expression was observed in respiratory brainstem neurons (putative ventral respiratory column) on the side of the injury, despite these molecules being involved in synaptic plasticity and axonal regeneration [56].

The present study also demonstrated no significant change in GAP-43 expression within white matter regions with treatment. This raises the question as to whether neuroplasticity is being mediated by spinal networks, which many recruit spinal interneurons within the phrenic network. This would be consistent with injured mice [57] and rats [58,59] displaying increased connectivity of spinal interneurons after cervical spinal cord injury. Furthermore, stimulating activity within spinal networks has also been shown to increase spinal interneuron activity and plasticity [60]. Based on these prior results, it is likely that stimulation with rTMS may also drive a degree of plasticity within these same phrenic interneuronal networks. This could also explain why phrenic motor activity on the side of injury was increased during asphyxia 2 months after chronic rTMS. Indeed, this observation could be an indication of strengthening of the CPP involving both spared descending pathways and interneuronal connectivity.

In this study, we chose to specifically stimulate at the cortical level to target neuronal networks connected to brainstem respiratory centers. Previous studies using repetitive magnetic stimulation employed a thoracic model of SCI, applying magnetic stimulation at the side of injury. Studies in mice demonstrated that 10 Hz acute or chronic rTSMS improved locomotor function after thoracic injury [45,61], but a lower frequency (0.2 Hz) of rTSMS in rats did not [40]. However, when combining stimulation with growth factor delivery and activity-based therapy, improved functional effects were observed [40].

More invasive spinal stimulation approaches, such as intraspinal and epidural stimulation, have also been used to elicit spinal neural plasticity [62,63,64,65]. High-frequency epidural stimulation was used at the level of the phrenic motor nucleus (C4 segment) in C1-transected animals to induce short-term facilitation of the phrenic motor circuit [64]. This has been demonstrated within the denervated phrenic motor pool after C2HS in rats with increased growth factor expression (e.g., VEGF and BDNF) [66].

## 5. Conclusions

Even though epidural stimulation is a promising technique, it is surgically invasive, and accordingly, less invasive approaches than that used in the present study may be more readily applied in the clinic. Despite some promise with TMS, the results from the present study may reflect the fact that 10 Hz rTMS alone may be insufficient to stimulate significant functional diaphragmatic recovery. Combining the rTMS protocol employed in the present study with other therapeutic interventions (e.g., activity-based therapy [40,67]) may be more efficacious.

## Figures and Tables

**Figure 1 biology-11-00473-f001:**
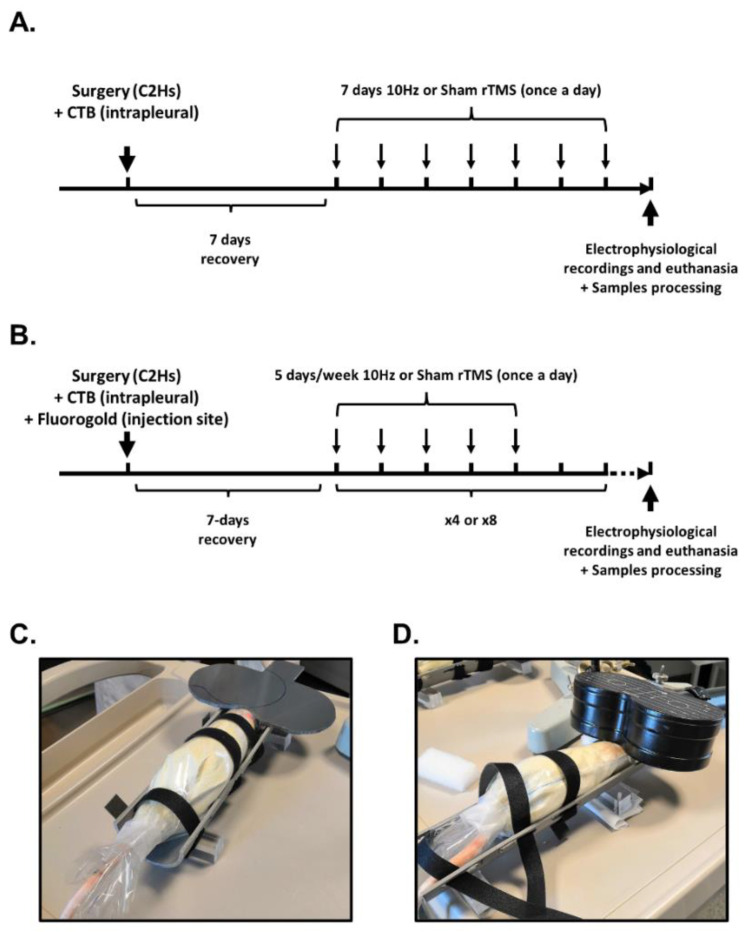
Protocols for 10 Hz or Sham rTMS following C2 spinal cord hemisection: (**A**) rTMS protocols for 7-day-treated groups; (**B**) rTMS protocols for 1-month- and 2-month-treated groups; (**C**) image of rat receiving Sham rTMS protocol; (**D**) image of rat receiving 10 Hz rTMS protocol.

**Figure 2 biology-11-00473-f002:**
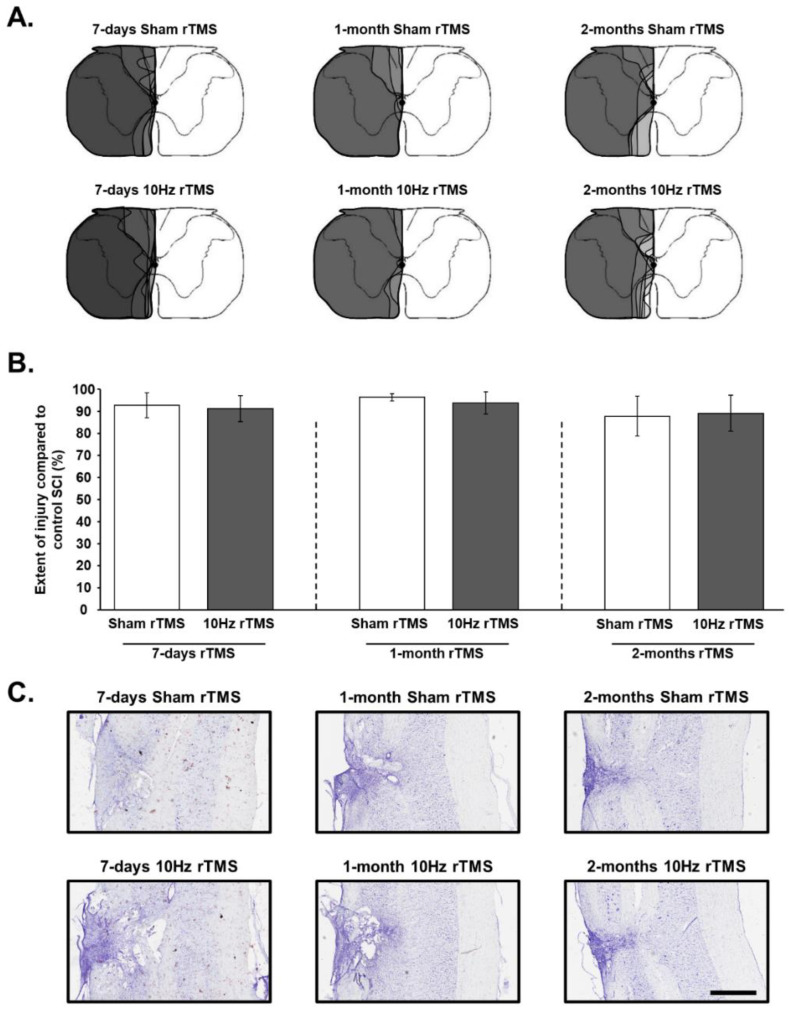
Extent of injury following a C2 spinal cord hemisection: (**A**) representative schematic diagrams of the extent of injury in each animal at 15 days postinjury (P.I.) for 7-day Sham and 10 Hz rTMS groups, 36 days P.I. for 1-month Sham and 10 Hz rTMS groups, and 64 days P.I. for 2-month Sham and 10 Hz rTMS groups; (**B**) extent of injury quantification in percentage compared with control spinal cord injury (SCI) 100%. The quantification has been made only in the ventral part where the phrenic motoneurons are located. There is no difference between the different groups (One Way ANOVA, *p* = 0.235); (**C**) representative image for each group stained in cresyl violet. Scale Bar: 1 mm.

**Figure 3 biology-11-00473-f003:**
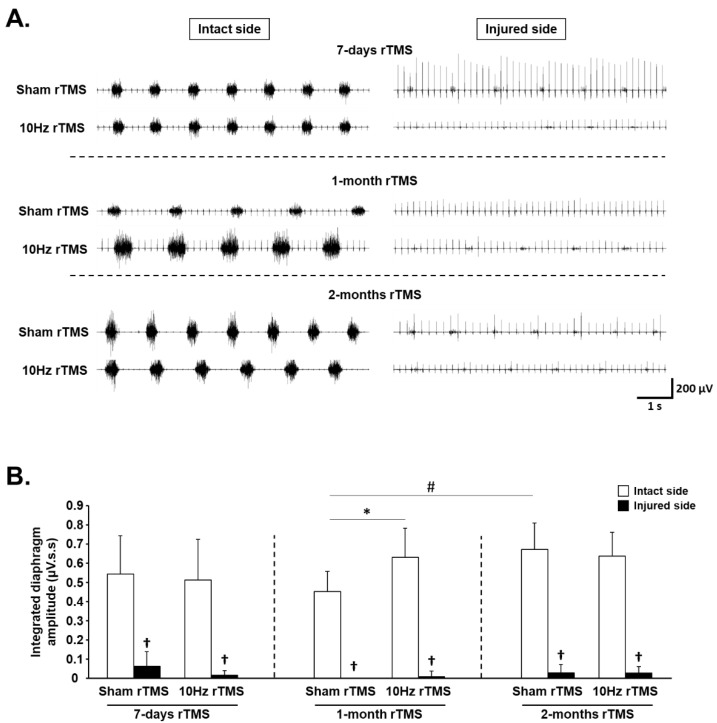
Diaphragm activity in C2 hemisected rats following chronic Sham and 10 Hz rTMS: (**A**) representative traces of raw diaphragm EMG of C2 hemisected rats, following 7-day, 1-month, or 2-month Sham or 10 Hz rTMS treatment; (**B**) integrated diaphragm amplitude for intact and injured sides of Sham or 10 Hz rTMS treated C2 hemisected animals following 7 days, 1 month, or 2 months of treatment. † *p* < 0.001, compared with intact side; * *p* = 0.011, intact side of Sham rTMS group vs. intact side of 10 Hz rTMS group following 1-month treatment. # *p* = 0.039, intact side of Sham rTMS group following 1-month treatment vs. corresponding group following 2 months of treatment.

**Figure 4 biology-11-00473-f004:**
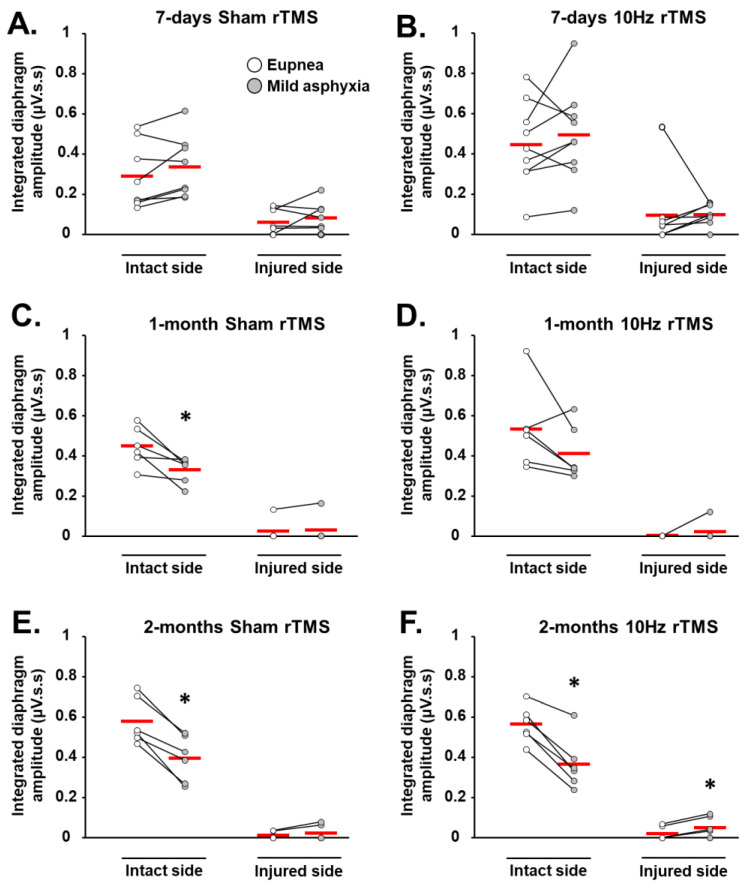
Diaphragm activity in C2 hemisected rats following chronic Sham and 10 Hz rTMS during respiratory challenge: integrated diaphragm amplitude for (**A**) 7-day Sham, (**B**) 7-day 10 Hz rTMS, (**C**) 1-month Sham, (**D**) 1-month 10 Hz rTMS, (**E**) 2-month Sham, and (**F**) 2-month 10 Hz rTMS groups in eupnea and during mild asphyxia. * *p* < 0.05 mild asphyxia, compared with eupnea (paired *t*-test). The red short line represent the mean value.

**Figure 5 biology-11-00473-f005:**
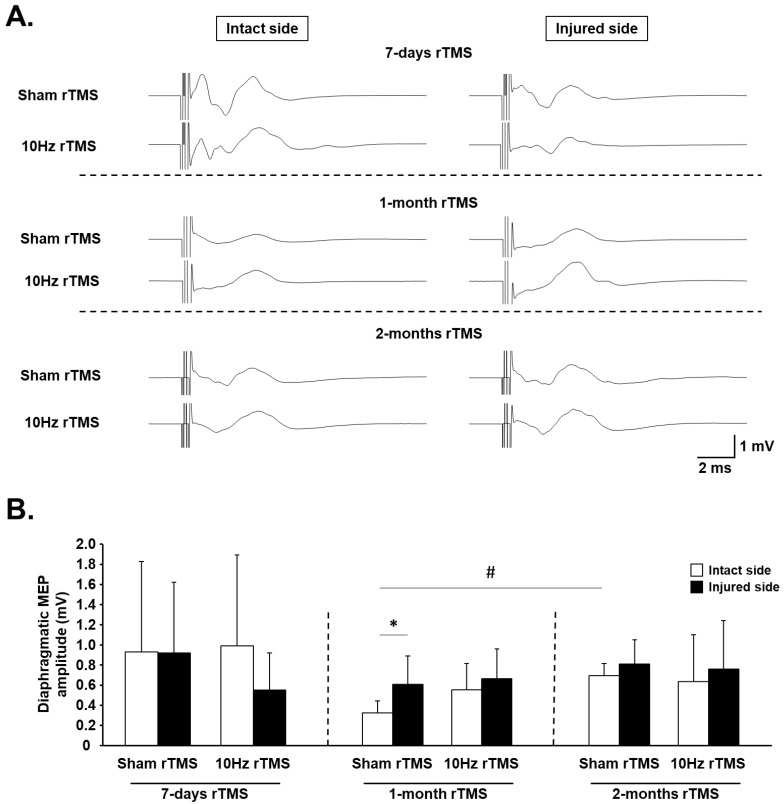
Diaphragm excitability in C2 hemisected rats following chronic Sham and 10 Hz rTMS: (**A**) representative traces of raw diaphragm MEP of C2 hemisected rats, following 7 days, 1 month, or 2 months of Sham or 10 Hz rTMS treatment; (**B**) MEP amplitude for intact and injured sides of Sham- or 10 Hz rTMS-treated C2 hemisected animals following 7 days, 1 month, or 2 months of treatment. * *p* = 0.028, intact side vs. injured side of Sham rTMS group following 1-month treatment. # *p* < 0.001, intact side of Sham rTMS group following 1-month treatment vs. corresponding group following 2 months of treatment.

**Figure 6 biology-11-00473-f006:**
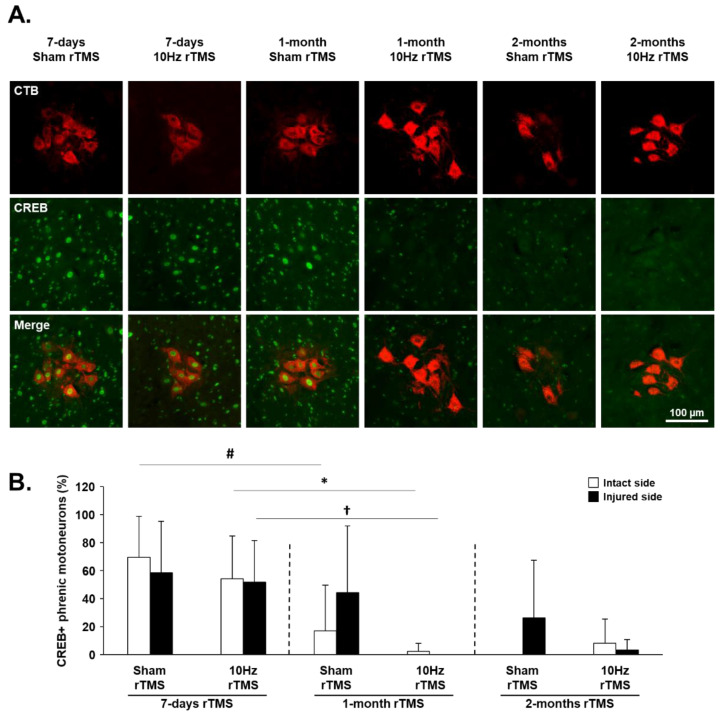
CREB expression in phrenic motoneurons following C2 hemisection: (**A**) representative images showing expression of CREB in denervated phrenic motoneurons labeled with CTB in C2 hemisected rats, following 7 days, 1 month or 2 months of Sham or 10 Hz rTMS treatment; (**B**) quantification of the percentage of CREB expressing phrenic motoneurons for intact and injured sides of Sham- or 10 Hz rTMS-treated C2 hemisected animals following 7 days, 1 month, or 2 months of treatment. There is no difference between the intact and the injured sides for the different groups (paired *t*-test for intact vs. injured side, *p* > 0.05); # 7-day Sham rTMS intact side, compared with 1-month Sham rTMS intact side, *p* = 0.012. * 7-day 10 Hz rTMS intact side, compared with 1-month 10 Hz rTMS intact side, *p* = 0.007. † 7-day 10 Hz rTMS injured side, compared with 1-month 10 Hz rTMS injured side, *p* = 0.004.

**Figure 7 biology-11-00473-f007:**
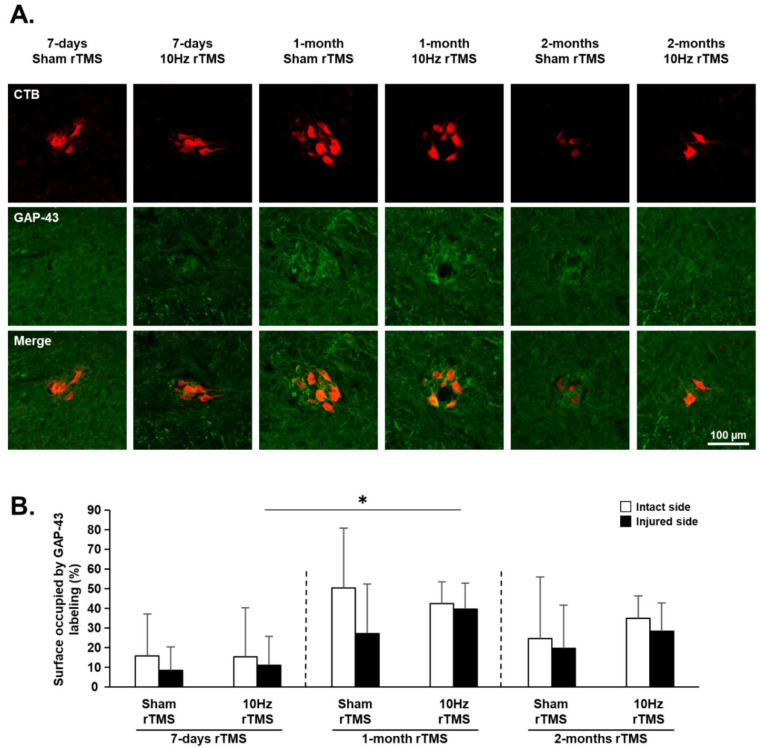
GAP-43 expression approximate to phrenic motoneurons following C2 hemisection: (**A**) representative images showing the surface occupied by GAP-43 labeling around denervated phrenic motoneurons labeled with CTB in C2 hemisected rats, following 7 days, 1 month, or 2 months of Sham or 10 Hz rTMS treatment; (**B**) quantification of the area occupied by GAP-43 labeling approximate to phrenic motoneurons on the intact and injured sides of Sham or 10 Hz rTMS-treated C2 hemisected animals following 7 days, 1 month, or 2 months of treatment. There is no difference between the intact and the injured sides for the different groups (paired *t*-test for intact vs. injured side, *p* > 0.05; * 7-day 10 Hz rTMS compared with 1-month 10 Hz rTMS for the injured side (Student’s *t*-test, *p* = 0.017).

## Data Availability

The data presented in this study are available on request from the corresponding author.

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
