# Peer review of "Effects of Chronic High-Frequency rTMS Protocol on Respiratory Neuroplasticity Following C2 Spinal Cord Hemisection in Rats"

_biology, 2022, doi:10.3390/biology11030473_

Round 1
Reviewer 1 Report
In this study, high frequency (10 Hz) rTMS was used to stimulate the C2 hemisected area of spinal cord for 7 days, 1 month, and 2 months to demonstrate the improvement of impaired diaphragmatic activities and its therapeutic effects on the inflammatory response of the injured spinal cord. However, the authors need to address some issues in order to improve the experimental results presented in this study.
- In Figure 4&5, the authors should provide immunofluorescence images of the entire spinal cord section, including the injured and intact sides, not just a few cells.
- According to the results in Figure 4, CREB was significantly reduced in the spinal cord of the injured side after 1-month and 2-month 10Hz rTMS treatment. However, the results in the figure also showed that after two months of sham rTMS treatment, the expression of CREB in the spinal cord also decreased significantly. How can this be explained? Is sham rTMS also effective?
- Images and data of Iba1 and CD68 expression in Figure 6 were missing after 7-days sham or 10Hz rTMS treatment.
- The data and images in Figure 7 have the same problem as above.
- The immunofluorescence staining results of Figure 7 are too blurry to be seen clearly.
- The FG fluorescence in Supplementary Figures 6 and 7 are exactly the same, please explain how CREB and WFA were stained with the same slice, and the latter two are both red fluorescents.
Author Response
Reviewer 1 :
In this study, high frequency (10 Hz) rTMS was used to stimulate the C2 hemisected area of spinal cord for 7 days, 1 month, and 2 months to demonstrate the improvement of impaired diaphragmatic activities and its therapeutic effects on the inflammatory response of the injured spinal cord. However, the authors need to address some issues in order to improve the experimental results presented in this study.
- In Figure 4&5, the authors should provide immunofluorescence images of the entire spinal cord section, including the injured and intact sides, not just a few cells.
We have now included in a supplemental figure the entire spinal cord for CTB/GAP43 and CTB/Creb immunostainings.
- According to the results in Figure 4, CREB was significantly reduced in the spinal cord of the injured side after 1-month and 2-month 10Hz rTMS treatment. However, the results in the figure also showed that after two months of sham rTMS treatment, the expression of CREB in the spinal cord also decreased significantly. How can this be explained? Is sham rTMS also effective?
The CREB expression reduction is accelerated with the 10Hz rTMS treatment compared to Sham treated animals. Our treatment accelerate the CREB expression decrease. We already discussed about this reduction in the discussion section : « The reduction of CREB expression in phrenic motoneurons occurred earlier post-injury in treated animals, suggesting that CREB signaling may contribute to treatment-driven plasticity. »
- Images and data of Iba1 and CD68 expression in Figure 6 were missing after 7-days sham or 10Hz rTMS treatment.
We were unable to run Iba1 and CD68 immunolabeling on spinal tissue from animals 7 days post-rTMS or sham treatment because the tissue was used for another study, and, unfortunately we did not have enough slides left for the present study.
- The data and images in Figure 7 have the same problem as above.
Same comment as before.
- The immunofluorescence staining results of Figure 7 are too blurry to be seen clearly.
For the final version of the manuscript before publication, we will make sure that our figure quality is high enough to avoid such blurriness.
- The FG fluorescence in Supplementary Figures 6 and 7 are exactly the same, please explain how CREB and WFA were stained with the same slice, and the latter two are both red fluorescents.
Indeed, that is the same slice for FG staining. When we did the IF, we did FG, CREB (secondary antibody Alexa 647) and WFA (Avidin 488) on brainstem slice. For better contrasts, we choose another red color, since our camera pick up only grayscale values, and the rendering colors are de facto « fake » colors, and we choose red. In the material and method section, we did mentionned the antibodies we used, and obviously, what we show it is only false colors.
Reviewer 2 Report
“Effects of chronic high-frequency rTMS protocol on respiratory neuroplasticity following C2 spinal cord hemisection in the rat”
Overall strengths of the article:
This manuscript aimed to test the potential therapeutic effects of high-frequency rTMS on impaired respiratory function following cervical SCI. The authors have tested the hypothesis that chronic 10Hz rTMS can improve respiratory function after SCI. They tested this hypothesis using cellular, molecular and electrophysiological outcome measures in a preclinical model of C2- spinal hemisection injury in adult rats. They demonstrated that increase of intact hemidiaphragm electromyogram (EMG) activity and excitability (diaphragm motor evoked potentials) after 1-month of rTMS application but not after chronic application. Interestingly, despite no real functional effects of rTMS treatment on the injured hemidiaphragm activity during eupnea, 2-months of rTMS treatment strengthened the existing crossed phrenic pathways, allowing the injured hemidiaphragm to increase its activity during the respiratory challenge (i.e. asphyxia).
This study addresses an important question in the recovery of impaired respiratory function following a high cervical SCI, a long-standing problem for which no satisfactory solution has been found yet. An interesting study, considering the non-invasive nature of treatment that can be easily translated to the clinic. However, this paper suffers from a number of major limitations that should be addressed before publication. Details in the specific comments section.
Specific comments on weaknesses:
Major concerns:
- Can the authors show the anatomy of the C2 spinal cord lesion? How reproducible was it?
- Is neuronal survival affected after rTMS treatment?
- “Non-treated animals reached an EMGdia amplitude similar to those of treated animals at 9 weeks post-injury. These results might suggest that chronic 10Hz rTMS strengthens spared descending respiratory pathways, reflecting a recovery plateau”. Can the author elaborate more on the recovery curve? I strongly suggest discussing it in more detail with respective citations.
- Methods need to be presented in a better way; especially "section 2.2. Chronic C2 hemisection" and "2.4. Electrophysiological recordings". Multiple procedures can be described in may be sub-section to make it more clear.
- I strongly recommend presenting supplementary figures 1 and 2 as the main figure in the main manuscript, it will help a lot to understand the methods in detail, and you can move figures 6 & 7 to the supplementary materials.
Minor points:
- Single data points should be shown whenever possible in the graph.
- For the limited novel observations made, parts of the paper are too long and some figures can easily be omitted (can be described in a few sentences).
- Typos; ‘(rTMS))’; section 2.1. ‘Ethivs’ statement; discussion: pool after ‘C2Hs’, and others.
- English correction; “the results obtained in this present manuscript suggest”
Author Response
Reviewer 2 :
“Effects of chronic high-frequency rTMS protocol on respiratory neuroplasticity following C2 spinal cord hemisection in the rat”
Overall strengths of the article:
This manuscript aimed to test the potential therapeutic effects of high-frequency rTMS on impaired respiratory function following cervical SCI. The authors have tested the hypothesis that chronic 10Hz rTMS can improve respiratory function after SCI. They tested this hypothesis using cellular, molecular and electrophysiological outcome measures in a preclinical model of C2- spinal hemisection injury in adult rats. They demonstrated that increase of intact hemidiaphragm electromyogram (EMG) activity and excitability (diaphragm motor evoked potentials) after 1-month of rTMS application but not after chronic application. Interestingly, despite no real functional effects of rTMS treatment on the injured hemidiaphragm activity during eupnea, 2-months of rTMS treatment strengthened the existing crossed phrenic pathways, allowing the injured hemidiaphragm to increase its activity during the respiratory challenge (i.e. asphyxia).
We thank the reviewer for this positive comment.
This study addresses an important question in the recovery of impaired respiratory function following a high cervical SCI, a long-standing problem for which no satisfactory solution has been found yet. An interesting study, considering the non-invasive nature of treatment that can be easily translated to the clinic. However, this paper suffers from a number of major limitations that should be addressed before publication. Details in the specific comments section.
Specific comments on weaknesses:
Major concerns:
- Can the authors show the anatomy of the C2 spinal cord lesion? How reproducible was it?
The reproducibility of our injuries is displayed in Supplemental Figure 2 in the version 1 of the manuscript. It is highly reproducible, and there is no statistical difference in term of injury extent between all the groups. We added representative histological pictures of the injury for each studied group.
- Is neuronal survival affected after rTMS treatment?
In this present study, we did not evaluate the neuronal survival after rTMS treatment around the injury site and even in the injured phrenic pre-motoneurons. However, in this particular model (C2 hemisection in rat), there is almost no neuronal death in the respiratory network affected by the initial injury (Vinit and Kastner, 2009 for review, Allen et al., 2021). We only deafferented the phrenic motoneurons, and this deafferentation does not induce neuronal death. Even in the brainstem injured neurons (in this study, we labeled it with fluorogold), the initial injury is far away from the neuronal body, and the neurons does not express any apopotosis markers, only regenerative associated proteins markers (Vinit et al., 2005, Darlot et al., 2017). Indeed, we did not check for other non-respiratory related neurons, and we will probably look at it in a future study.
- “Non-treated animals reached an EMGdia amplitude similar to those of treated animals at 9 weeks post-injury. These results might suggest that chronic 10Hz rTMS strengthens spared descending respiratory pathways, reflecting a recovery plateau”. Can the author elaborate more on the recovery curve? I strongly suggest discussing it in more detail with respective citations.
Thanks for the comment. We did more discussion about the spontaneous recovery of diaphragmatic function and made the change accordingly.
« Indeed, between 4 and 8 weeks post-C2 hemisection, a plateau in diaphragm activity has been observed when treated with intermittent hypoxia [53]. A similar observation was also found when a chronic protocol of intermittent hypoxia was applied following C2 hemisection in rat on diaphragm activity, with no difference between treated animals and normoxic animals after 3 weeks of treatment (spontaneous recovery reaches a ceiling/plateau by that time post-injury) [54] »
- Methods need to be presented in a better way; especially "section 2.2. Chronic C2 hemisection" and "2.4. Electrophysiological recordings". Multiple procedures can be described in may be sub-section to make it more clear.
Thanks for the comment, we made the change accordingly.
- I strongly recommend presenting supplementary figures 1 and 2 as the main figure in the main manuscript, it will help a lot to understand the methods in detail, and you can move figures 6 & 7 to the supplementary materials.
We did moved the figure 6 and 7 to the supplementary data, and the supplemental figure 1 and 2 to the manuscript’s figure.
Minor points:
- Single data points should be shown whenever possible in the graph.
Thanks you for this comment. We already have a figure with single data points in figure 2 (which is interesting to follow the same animal overtime), but for the other data, the visualization of the graph will be harder.
- For the limited novel observations made, parts of the paper are too long and some figures can easily be omitted (can be described in a few sentences).
We agree with the reviewer and we moved 2 figures (6 and 7) into the supplementary data.
- Typos; ‘(rTMS))’; section 2.1. ‘Ethivs’ statement; discussion: pool after ‘C2Hs’, and others.
We did the change accordingly.
- English correction; “the results obtained in this present manuscript suggest”
We did the change accordingly.
Round 2
Reviewer 1 Report
The authors addressed my questions in an almost satisfactory way. But there is still a minor question:
Since you cannot provide data for rTMS treatment 7 days after C2 lesions, it should not be mentioned in Supplementary Figure 7 "Representative images showing expression of Iba1 and CD68 at the site of the lesion in C2 hemisected rats, following 7-days, 1-month or 2-months Sham or 10Hz rTMS treatment”, please rewrite the sentence description.
Author Response
The authors addressed my questions in an almost satisfactory way. But there is still a minor question:
We thank you for this positive comment.
Since you cannot provide data for rTMS treatment 7 days after C2 lesions, it should not be mentioned in Supplementary Figure 7 "Representative images showing expression of Iba1 and CD68 at the site of the lesion in C2 hemisected rats, following 7-days, 1-month or 2-months Sham or 10Hz rTMS treatment”, please rewrite the sentence description.
We made the change accordingly in the legend of the Supplementary Figure 7 and 8 to match the reviewer comment.
Reviewer 2 Report
This study addresses an important question in the recovery of impaired respiratory function following high cervical SCI, a long-standing problem for which no satisfactory solution has been found yet. In the revised manuscript authors have successfully addressed all the comments raised by the reviewer and incorporated all the suggestions to improve the quality of the paper. I think this manuscript has been sufficiently improved from the previous version.
Author Response
This study addresses an important question in the recovery of impaired respiratory function following high cervical SCI, a long-standing problem for which no satisfactory solution has been found yet. In the revised manuscript authors have successfully addressed all the comments raised by the reviewer and incorporated all the suggestions to improve the quality of the paper. I think this manuscript has been sufficiently improved from the previous version.
We thank the reviewer for this postive comment.